# Fast greedy algorithms for dictionary selection with generalized sparsity constraints

**Kaito Fujii**
Graduate School of Information Sciences and Technology
The University of Tokyo
kaito_fujii@mist.i.u-tokyo.ac.jp

**Tasuku Soma**
Graduate School of Information Sciences and Technology
The University of Tokyo
tasuku_soma@mist.i.u-tokyo.ac.jp

## Abstract

In dictionary selection, several atoms are selected from finite candidates that successfully approximate given data points in the sparse representation. We propose a novel efficient greedy algorithm for dictionary selection. Not only does our algorithm work much faster than the known methods, but it can also handle more complex sparsity constraints, such as average sparsity. Using numerical experiments, we show that our algorithm outperforms the known methods for dictionary selection, achieving competitive performances with dictionary learning algorithms in a smaller running time.

## 1   Introduction

Learning sparse representations of data and signals has been extensively studied for the past decades in machine learning and signal processing [16]. In these methods, a specific set of basis signals (atoms), called a *dictionary*, is required and used to approximate a given signal in a sparse representation. The design of a dictionary is highly nontrivial, and many studies have been devoted to the construction of a good dictionary for each signal domain, such as natural images and sounds. Recently, approaches to construct a dictionary from data have shown the state-of-the-art results in various domains. The standard approach is called *dictionary learning* [3, 32, 1]. Although many studies have been devoted to dictionary learning, it is usually difficult to solve, requiring a non-convex optimization problem that often suffers from local minima. Also, standard dictionary learning methods (e.g., MOD [14] or $k$-SVD [2]) require a heavy time complexity.

Krause and Cevher [22] proposed a combinatorial analogue of dictionary learning, called *dictionary selection*. In dictionary selection, given a finite set of candidate atoms, a dictionary is constructed by selecting a few atoms from the set. Dictionary selection could be faster than dictionary learning due to its discrete nature. Another advantage of dictionary selection is that the approximation guarantees hold even in agnostic settings, i.e., we do not need stochastic generating models of the data. Furthermore, dictionary selection algorithms can be used for *media summarization*, in which the atoms must be selected from given data points [8, 9].

The basic dictionary selection is formalized as follows. Let $V$ be a finite set of candidate atoms and $n = |V|$. Throughout the paper, we assume that the atoms are unit vectors in $\mathbb{R}^d$ without loss of generality. We represent the candidate atoms as a matrix $\mathbf{A} \in \mathbb{R}^{d \times n}$ whose columns are the atoms in $V$. Let $\mathbf{y}_t \in \mathbb{R}^d$ ($t \in [T]$) be data points, where $[T] = \{1, \dots, T\}$, and $k$ and $s$ be positive integers

with $k \geq s$. We assume that a utility function $u : \mathbb{R}^d \times \mathbb{R}^d \to \mathbb{R}_+$ exists, which measures the similarity of the input vectors. For example, one can use the $\ell^2$-utility function $u(\mathbf{y}, \mathbf{x}) = \|\mathbf{y}\|_2^2 - \|\mathbf{y} - \mathbf{x}\|_2^2$ as in Krause and Cevher [22]. Then, the dictionary selection finds a set $X \subseteq V$ of size $k$ that maximizes

$$h(X) = \sum_{t=1}^{T} \max_{\mathbf{w} \in \mathbb{R}^k : \|\mathbf{w}\|_0 \leq s} u(\mathbf{y}_t, \mathbf{A}_X \mathbf{w}), \tag{1}$$

where $\|\mathbf{w}\|_0$ is the number of nonzero entries in $\mathbf{w}$ and $\mathbf{A}_X$ is the column submatrix of $\mathbf{A}$ with respect to $X$. That is, we approximate a data point $\mathbf{y}_t$ with a sparse representation in atoms in $X$, where the approximation quality is measured by $u$. Letting $f_t(Z_t) := \max_{\mathbf{w}} u(\mathbf{y}_t, \mathbf{A}_{Z_t} \mathbf{w})$ $(t \in [T])$, we can rewrite this as the following *two-stage optimization*: $h(X) = \sum_{t=1}^{T} \max_{Z_t \subseteq X : |Z_t| \leq s} f_t(Z_t)$. Here $Z_t$ is the set of atoms used in a sparse representation of data point $\mathbf{y}_t$. The main challenges in dictionary selection are that the evaluation of $h$ is NP-hard in general [25], and the objective function $h$ is not submodular [17] and therefore the well-known greedy algorithm [27] cannot be applied. The previous approaches construct a good proxy of dictionary selection that can be easily solved, and analyze the approximation ratio.

## 1.1 Our contribution

Our main contribution is a novel and efficient algorithm called the *replacement orthogonal matching pursuit (Replacement OMP)* for dictionary selection. This algorithm is based on a previous approach called Replacement Greedy [30] for *two-stage submodular maximization*, a similar problem to dictionary selection. However, the algorithm was not analyzed for dictionary selection. We extend their approach to dictionary selection in the present work, with an additional improvement that exploits techniques in orthogonal matching pursuit. We compare our method with the previous methods in Table 1. Replacement OMP has a smaller running time than SDS$_{\text{OMP}}$ [10] and Replacement Greedy. The only exception is SDS$_{\text{MA}}$ [10], which intuitively ignores any correlation of the atoms. In our experiment, we demonstrate that Replacement OMP outperforms SDS$_{\text{MA}}$ in terms of test residual variance. We note that the constant approximation ratios of SDS$_{\text{MA}}$, Replacement Greedy, and Replacement OMP are incomparable in general. In addition, we demonstrate that Replacement OMP achieves a competitive performance with dictionary learning algorithms in a smaller running time, in numerical experiments.

**Generalized sparsity constraint**  Incorporating further prior knowledge on the data domain often improves the quality of dictionaries [28, 29, 11]. A typical example is a combinatorial constraint independently imposed on each support $Z_t$. This can be regarded as a natural extension of the *structured sparsity* [19] in sparse regression, which requires the support to satisfy some combinatorial constraint, rather than a cardinality constraint. A *global structure* of supports is also useful prior information. Cevher and Krause [6] proposed a global sparsity constraint called the *average sparsity*, in which they add a global constraint $\sum_{t=1}^{T} |Z_t| \leq s'$. Intuitively, the average sparsity constraint requires that the most data points can be represented by a small number of atoms. If the data points are patches of a natural image, most patches are a simple background, and therefore the number of the total size of the supports must be small. The average sparsity has been also intensively studied in dictionary learning [11]. To deal with these generalized sparsities in a unified manner, we propose a novel class of sparsity constraints, namely *p-replacement sparsity families*. We prove that Replacement OMP can be applied for the generalized sparsity constraint with a slightly worse approximation ratio. We emphasize that the OMP approach is essential for *efficiency*; in contrast, Replacement Greedy cannot be extended to the average sparsity setting because it can only handle local constraints on $Z_t$, and yields an exponential running time.

**Online extension**  In some practical situations, it is not always feasible to store all data points $\mathbf{y}_t$, but these data points arrive in an online fashion. We show that Replacement OMP can be extended to the online setting, with a sublinear approximate regret. The details are given in Section 5.

## 1.2 Related work

Krause and Cevher [22] first introduced dictionary selection as a combinatorial analogue of dictionary learning. They proposed SDS$_{\text{MA}}$ and SDS$_{\text{OMP}}$, and analyzed the approximation ratio using the *coherence* of the matrix $\mathbf{A}$. Das and Kempe [10] introduced the concept of the *submodularity ratio*

Table 1: Comparison of known methods with Replacement OMP. The constants $m_s$, $M_s$, and $M_{s,2}$ are the restricted concavity and smoothness constants of $u(\mathbf{y}_t, \cdot)$ ($t \in [T]$); see Section 2. The running time is from the $\ell^2$-utility function $u$ and the individual sparsity constraint.

| Method | Approximation ratio | Running time | Generalized sparsity |
|---|---|---|---|
| SDS$_{\mathrm{MA}}$ [22] | $\frac{m_1 m_s}{M_1 M_s}(1 - 1/e)$ [10] | $\mathrm{O}((k+d)nT)$ | No |
| SDS$_{\mathrm{OMP}}$ [22] | $\mathrm{O}(1/k)$ [10] | $\mathrm{O}((s+k)sdknT)$ | No |
| Replacement Greedy [30] | $\left(\frac{m_{2s}}{M_{s,2}}\right)^2\left(1 - \exp\left(-\frac{M_{s,2}}{m_{2s}}\right)\right)$ | $\mathrm{O}(s^2dknT)$ | No |
| Replacement OMP | $\left(\frac{m_{2s}}{M_{s,2}}\right)^2\left(1 - \exp\left(-\frac{M_{s,2}}{m_{2s}}\right)\right)$ | $\mathrm{O}((n+ds)kT)$ | Yes |

and refined the analysis via the *restricted isometry property* [5]. A connection to the restricted concavity and submodularity ratio has been investigated by Elenberg et al. [13], Khanna et al. [21] for sparse regression and matrix completion. Balkanski et al. [4] studied two-stage submodular maximization as a submodular proxy of dictionary selection, devising various algorithms. Stan et al. [30] proposed Replacement Greedy for two-stage submodular maximization. It is unclear that these methods provide an approximation guarantee for the original dictionary selection.

To the best of our knowledge, there is no existing research in the literature that addresses online dictionary selection. For a related problem in sparse optimization, namely *online linear regression*, Kale et al. [20] proposed an algorithm based on *supermodular minimization* [23] with a sublinear approximate regret guarantee. Elenberg et al. [12] devised a streaming algorithm for weak submodular function maximization. Chen et al. [7] dealt with online maximization of weakly DR-submodular functions.

**Organization**   The rest of this paper is organized as follows. Section 2 provides the basic concepts and definitions. Section 3 formally defines dictionary selection with generalized sparsity constraints. Section 4 presents our algorithm, Replacement OMP. Section 5 sketches the extension to the online setting. The experimental results are presented in Section 6.

## 2   Preliminaries

**Notation**   For a positive integer $n$, $[n]$ denotes the set $\{1, 2, \ldots, n\}$. The sets of reals and nonnegative reals are denoted by $\mathbb{R}$ and $\mathbb{R}_{\geq 0}$, respectively. We similarly define $\mathbb{Z}$ and $\mathbb{Z}_{\geq 0}$. Vectors and matrices are denoted by lower and upper case letters in boldface, respectively: $\mathbf{a}, \mathbf{x}, \mathbf{y}$ for vectors and $\mathbf{A}, \mathbf{X}, \mathbf{Y}$ for matrices. The $i$th standard unit vector is denoted by $\mathbf{e}_i$; that is, $\mathbf{e}_i$ is the vector such that its $i$th entry is equal to one and all other entries are zero. For a matrix $\mathbf{A} \in \mathbb{R}^{d \times n}$ and $X \subseteq [n]$, $\mathbf{A}_X$ denotes the column submatrix of $\mathbf{A}$ with respect to $X$. The maximum and minimum singular values of a matrix $\mathbf{A}$ are denoted by $\sigma_{\max}(\mathbf{A})$ and $\sigma_{\min}(\mathbf{A})$, respectively. For a positive integer $k$, we define $\sigma_{\max}(\mathbf{A}, k) := \max_{X \subseteq [n]: |X| \leq k} \sigma_{\max}(\mathbf{A}_X)$. We define $\sigma_{\min}(\mathbf{A}, k)$ in a similar way. For $t \in [T]$, let $u_t(\mathbf{w}) := u(\mathbf{y}_t, \mathbf{A}\mathbf{w})$. Let $\mathbf{w}_t^{(Z_t)}$ denote the maximizer of $u_t(\mathbf{w})$ subject to $\mathrm{supp}(\mathbf{w}) \subseteq Z_t$. Throughout the paper, $V$ denotes the fixed finite ground set. For $X \subseteq V$ and $a \in V \setminus X$, we define $X + a := X \cup \{a\}$. Similarly, for $a \in V \setminus X$ and $b \in X$, we define $X - b + a := (X \setminus \{b\}) \cup \{a\}$.

### 2.1   Restricted concavity and smoothness

The following concept of restricted strong concavity and smoothness is crucial in our analysis.

**Definition 2.1** (Restricted strong concavity and restricted smoothness [26]). Let $\Omega$ be a subset of $\mathbb{R}^d \times \mathbb{R}^d$ and $u \colon \mathbb{R}^d \to \mathbb{R}$ be a continuously differentiable function. We say that $u$ is *restricted strongly concave* with parameter $m_\Omega$ and *restricted smooth* with parameter $M_\Omega$ if,

$$-\frac{m_\Omega}{2}\|\mathbf{y} - \mathbf{x}\|_2^2 \geq u(\mathbf{y}) - u(\mathbf{x}) - \langle \nabla u(\mathbf{x}), \mathbf{y} - \mathbf{x}\rangle \geq -\frac{M_\Omega}{2}\|\mathbf{y} - \mathbf{x}\|_2^2$$

for all $(\mathbf{x}, \mathbf{y}) \in \Omega$.

We define $\Omega_{s,p} := \{(\mathbf{x}, \mathbf{y}) \in \mathbb{R}^d \times \mathbb{R}^d \colon \|\mathbf{x}\|_0, \|\mathbf{y}\|_0 \leq s, \|\mathbf{x} - \mathbf{y}\|_0 \leq p\}$ and $\Omega_s := \Omega_{s,s}$ for positive integers $s$ and $p$. We often abbreviate $M_{\Omega_s}$, $M_{\Omega_{s,p}}$, and $m_{\Omega_s}$ as $M_s$, $M_{s,p}$, and $m_s$, respectively.

# 3 Dictionary selection with generalized sparsity constraints

In this section, we formalize our problem, *dictionary selection with generalized sparsity constraints*. In this setting, the supports $Z_t$ for each $t \in [T]$ cannot be independently selected, but we impose a global constraint on them. We formally write such constraints as a down-closed [1] family $\mathcal{I} \subseteq \prod_{t=1}^{T} 2^V$. Therefore, we aim to find $X \subseteq V$ with $|X| \leq k$ maximizing

$$h(X) = \max_{Z_1,\ldots,Z_t \subseteq X \,:\, (Z_1,\ldots,Z_t) \in \mathcal{I}} \sum_{t=1}^{T} f_t(Z_t) \qquad (2)$$

Since a general down-closed family is too abstract, we focus on the following class. First, we define the set of *feasible replacements* for the current support $Z_1, \cdots, Z_T$ and an atom $a$ as

$$\mathcal{F}_a(Z_1, \cdots, Z_T) = \{(Z'_1, \cdots, Z'_T) \in \mathcal{I} : Z'_t \subseteq Z_t + a,\ |Z_t \setminus Z'_t| \leq 1\ (\forall t \in [T])\}. \qquad (3)$$

That is, the set of members in $\mathcal{I}$ obtained by adding $a$ and removing at most one element from each $Z_t$. Let $\mathcal{F}(Z_1, \cdots, Z_T) = \bigcup_{a \in V} \mathcal{F}_a(Z_1, \cdots, Z_T)$. If $Z_1, \ldots, Z_T$ are clear from the context, we simply write it as $\mathcal{F}_a$.

**Definition 3.1** (*p*-replacement sparsity). A sparsity constraint $\mathcal{I} \subseteq \prod_{t=1}^{T} 2^V$ is *p-replacement sparse* if for any $(Z_1, \ldots, Z_T), (Z_1^*, \ldots, Z_T^*) \in \mathcal{I}$, there is a sequence of $p$ feasible replacements $(Z_1^{p'}, \ldots, Z_T^{p'}) \in \mathcal{F}(Z_1, \ldots, Z_T)$ $(p' \in [p])$ such that each element in $Z_t^* \setminus Z_t$ appears at least once in the sequence $(Z_t^{p'} \setminus Z_t)_{p'=1}^{p}$ and each element in $Z_t \setminus Z_t^*$ appears at most once in the sequence $(Z_t \setminus Z_t^{p'})_{p'=1}^{p}$.

The following sparsity constraints are all *p*-replacement sparsity families. See Appendix B for proof.

**Example 3.2** (individual sparsity). The sparsity constraint for the standard dictionary selection can be written as $\mathcal{I} = \{(Z_1, \cdots, Z_T) \mid |Z_t| \leq s\ (\forall t \in [T])\}$. We call it *the individual sparsity constraint*. This constraint is a special case of an individual matroid constraint, described below.

**Example 3.3** (individual matroids). This was proposed by [30] as a sparsity constraint for two-stage submodular maximization. An *individual matroid constraint* can be written as $\mathcal{I} = \{(Z_1, \cdots, Z_T) \mid Z_t \in \mathcal{I}_t\ (\forall t \in [T])\}$ where $(V, \mathcal{I}_t)$ is a matroid [2] for each $t \in [T]$. An individual sparsity constraint is a special case of an individual matroid constraint where $(V, \mathcal{I}_t)$ is the uniform matroid for all $t$.

**Example 3.4** (block sparsity). Block sparsity was proposed by Krause and Cevher [22]. This sparsity requires that the support must be sparse within each prespecified block. That is, disjoint blocks $B_1, \cdots, B_b \subseteq [T]$ of data points are given in advance, and an only small subset of atoms can be used in each block. Formally, $\mathcal{I} = \{(Z_1, \cdots, Z_T) \mid |\bigcup_{t \in B_{b'}} Z_t| \leq s_{b'}\ (\forall b' \in [b])\}$ where $s_{b'} \in \mathbb{Z}_{\geq 0}$ for each $b' \in [b]$ are sparsity parameters.

**Example 3.5** (average sparsity [6]). This sparsity imposes a constraint on the average number of used atoms among all data points. The number of atoms used for each data point is also restricted. Formally, $\mathcal{I} = \{(Z_1, \cdots, Z_T) \mid |Z_t| \leq s_t, \sum_{t=1}^{T} |Z_t| \leq s'\}$ where $s_t \in \mathbb{Z}_{\geq 0}$ for each $t \in [T]$ and $s' \in \mathbb{Z}_{\geq 0}$ are sparsity parameters.

**Proposition 3.6.** *The replacement sparsity parameters of individual matroids, block sparsity, and average sparsity are upper-bounded by $k$, $k$, and $3k - 1$, respectively.*

# 4 Algortihms

In this section, we present Replacement Greedy [30] and Replacement OMP for dictionary selection with generalized sparsity constraints.

## 4.1 Replacement Greedy

Replacement Greedy was first proposed as an algorithm for a different problem, *two-stage submodular maximization* [4]. In two-stage submodular maximization, the goal is to maximize

$$h(X) = \sum_{t=1}^{T} \max_{Z_t \subseteq X \,:\, Z_t \in \mathcal{I}_t} f_t(Z_t), \tag{4}$$

where $f_t$ is a nonnegative monotone submodular function ($t \in [T]$) and $\mathcal{I}_t$ is a matroid. Despite the similarity of the formulation, in dictionary selection, the functions $f_t$ are not necessarily submodular, but come from the continuous function $u_t$. Furthermore, in two-stage submodular maximization, the constraints on $Z_t$ are individual for each $t \in [T]$, while we pose a global constraint $\mathcal{I}$. In the following, we present an adaptation of Replacement Greedy to dictionary selection with generalized sparsity constraints.

Replacement Greedy stores the current dictionary $X$ and supports $Z_t \subseteq X$ such that $(Z_1, \ldots, Z_T) \in \mathcal{I}$, which are initialized as $X = \varnothing$ and $Z_t = \varnothing$ ($t \in [T]$). At each step, the algorithm considers the gain of adding an element $a \in V$ to $X$ with respect to each function $f_t$, i.e., the algorithm selects $a$ that maximizes $\max_{(Z_1', \ldots, Z_T') \in \mathcal{F}_a} \sum_{t=1}^{T} \{f_t(Z_t') - f(Z_t)\}$. See Algorithm 1 for a pseudocode description. Note that for the individual matroid constraint $\mathcal{I}$, the algorithm coincides with the original Replacement Greedy [30].

---

**Algorithm 1** Replacement Greedy & Replacement OMP

---
1: Initialize $X \leftarrow \varnothing$ and $Z_t \leftarrow \varnothing$ for $t = 1, \ldots, T$.
2: **for** $i = 1, \ldots, k$ **do**
3:     Pick $a^* \in V$ that maximizes
$$\begin{cases} \max_{(Z_1', \cdots, Z_T') \in \mathcal{F}_{a^*}} \sum_{t=1}^{T} \{f_t(Z_t') - f_t(Z_t)\} & \text{(Replacement Greedy)} \\ \max_{(Z_1', \cdots, Z_T') \in \mathcal{F}_{a^*}} \left\{ \frac{1}{M_{s,2}} \sum_{t=1}^{T} \|\nabla u_t(\mathbf{w}_t^{(Z_t)})_{Z_t' \setminus Z_t}\|^2 - M_{s,2} \sum_{t=1}^{T} \|(\mathbf{w}_t^{(Z_t)})_{Z_t \setminus Z_t'}\|^2 \right\} \end{cases}$$
$$\text{(Replacement OMP)}$$
    and let $(Z_1', \cdots, Z_T')$ be a replacement achieving a maximum.
4:     Set $X \leftarrow X + a^*$ and $Z_t \leftarrow Z_t'$ for each $t \in [T]$.
5: **return** $X$.

---

Stan et al. [30] showed that Replacement Greedy achieves an $((1 - 1/\sqrt{e})/2)$-approximation when $f_t$ are monotone submodular. We extend their analysis to our non-submodular setting. The proof can be found in Appendix C.

**Theorem 4.1.** *Assume that $u_t$ is $m_{2s}$-strongly concave on $\Omega_{2s}$ and $M_{s,2}$-smooth on $\Omega_{s,2}$ for $t \in [T]$ and that the sparsity constraint $\mathcal{I}$ is $p$-replacement sparse. Let $(Z_1^*, \cdots, Z_T^*) \in \mathcal{I}$ be optimal supports of an optimal dictionary $X^*$. Then the solution $(Z_1, \cdots, Z_T) \in \mathcal{I}$ of Replacement Greedy after $k'$ steps satisfies*

$$\sum_{t=1}^{T} f_t(Z_t) \geq \frac{m_{2s}^2}{M_{s,2}^2} \left( 1 - \exp\left( -\frac{k'}{p} \frac{M_{s,2}}{m_{2s}} \right) \right) \sum_{t=1}^{T} f_t(Z_t^*)$$

## 4.2 Replacement OMP

Now we propose our algorithm, Replacement OMP. A down-side of Replacement Greedy is its heavy computation: in each greedy step, we need to evaluate $\sum_{t=1}^{T} f_t(Z_t')$ for each $(Z_1', \ldots, Z_t') \in \mathcal{F}_a(Z_1, \ldots, Z_t)$, which amounts to solving linear regression problems $snT$ times if $u$ is the $\ell^2$-utility function. To avoid heavy computation, we propose a proxy of this quantity by borrowing an idea from orthogonal matching pursuit. Replacement OMP selects an atom $a \in V$ that maximizes

$$\max_{(Z_1', \cdots, Z_T') \in \mathcal{F}_a(Z_1, \cdots, Z_T)} \left\{ \frac{1}{M_{s,2}} \sum_{t=1}^{T} \|\nabla u_t(\mathbf{w}_t^{(Z_t)})_{Z_t' \setminus Z_t}\|^2 - M_{s,2} \sum_{t=1}^{T} \|(\mathbf{w}_t^{(Z_t)})_{Z_t \setminus Z_t'}\|^2 \right\}. \tag{5}$$

This algorithm requires the smoothness parameter $M_{s,2}$ before the execution. Computing $M_{s,2}$ is generally difficult, but this parameter for the squared $\ell^2$-utility function can be bounded by $\sigma_{\max}^2(\mathbf{A}, 2)$. This value can be computed in $O(n^2 d)$ time.

**Theorem 4.2.** *Assume that $u_t$ is $m_{2s}$-strongly concave on $\Omega_{2s}$ and $M_{s,2}$-smooth on $\Omega_{s,2}$ for $t \in [T]$ and that the sparsity constraint $\mathcal{I}$ is $p$-replacement sparse. Let $(Z_1^*, \cdots, Z_T^*) \in \mathcal{I}$ be optimal supports of an optimal dictionary $X^*$. Then the solution $(Z_1, \cdots, Z_T) \in \mathcal{I}$ of Replacement OMP after $k'$ steps satisfies*

$$\sum_{t=1}^T f_t(Z_t) \geq \frac{m_{2s}^2}{M_{s,2}^2} \left( 1 - \exp\left( -\frac{k'}{p} \frac{M_{s,2}}{m_{2s}} \right) \right) \sum_{t=1}^T f_t(Z_t^*).$$

### 4.3 Complexity

Now we analyze the time complexity of Replacement Greedy and Replacement OMP. In general, $\mathcal{F}_a$ has $O(n^T)$ members, and therefore it is difficult to compute $\mathcal{F}_a$. Nevertheless, we show that Replacement OMP can run much faster for the examples presented in Section 3.

In Replacement Greedy, it is difficult to find an atom with the largest gain at each step. This is because we need to maximize a nonlinear function $\sum_{t=1}^T f_t(Z_t')$. Conversely, in Replacement OMP, if we can calculate $\mathbf{w}_t^{(Z_t)}$ and $\nabla u_t(\mathbf{w}_t^{(Z_t)})$ for all $t \in [T]$, the problem of calculating gain of each atom is reduced to maximizing a linear function.

In the following, we consider the $\ell_2$-utility function and average sparsity constraint because it is the most complex constraint. A similar result holds for the other examples. In fact, we show that this task reduces to maximum weighted bipartite matching. The Hungarian method returns the maximum weight bipartite matching in $O(T^3)$ time. We can further improve the running time to $O(T \log T)$ time by utilizing the structure of this problem. Due to the limitation of space, we defer the details to Appendix C. In summary, we obtain the following:

**Theorem 4.3.** *Assume that the assumption of Theorem 4.2 holds. Further assume that $u$ is the $\ell^2$-utility function and $\mathcal{I}$ is the average sparsity constraint. Then Replacement OMP finds the solution $(Z_1, \cdots, Z_T) \in \mathcal{I}$*

$$\sum_{t=1}^T f_t(Z_t) \geq \left( \frac{\sigma_{\max}^2(\mathbf{A}, 2s)}{\sigma_{\min}^2(\mathbf{A}, 2)} \right)^2 \left( 1 - \exp\left( -\frac{1}{3} \frac{\sigma_{\min}^2(\mathbf{A}, 2)}{\sigma_{\max}^2(\mathbf{A}, 2s)} \right) \right) \sum_{t=1}^T f_t(Z_t^*)$$

*in $O(Tk(n \log T + ds))$ time.*

**Remark 4.4.** If finding an atom with the largest gain is computationally intractable, we can add an atom whose gain is no less than $\tau$ times the largest gain. In this case, we can bound the approximation ratio with replacing $k'$ with $\tau k'$ in Theorem 4.1 and 4.2.

## 5 Extensions to the online setting

Our algorithms can be extended to the following online setting. The problem is formalized as a two-player game between a player and an adversary. At each round $t = 1, \ldots, T$, the player must select (possibly in a randomized manner) a dictionary $X_t \subseteq V$ with $|X_t| \leq k$. Then, the adversary reveals a data point $\mathbf{y}_t \in \mathbb{R}^d$ and the player gains $f_t(X_t) = \max_{\mathbf{w} \in \mathbb{R}^k : \|\mathbf{w}\|_0 \leq s} u(\mathbf{y}_t, \mathbf{A}_X \mathbf{w})$. The performance measure of a player's strategy is the *expected $\alpha$-regret*:

$$\text{regret}_\alpha(T) = \alpha \max_{X^* : |X^*| \leq k} \sum_{t=1}^T f_t(X^*) - \mathbf{E}\left[ \sum_{t=1}^T f_t(X_t) \right],$$

where $\alpha > 0$ is a constant independent from $T$ corresponding to the offline approximation ratio, and the expectation is taken over the randomness in the player.

For this online setting, we present an extension of Replacement Greedy and Replacement OMP with sublinear $\alpha$-regret, where $\alpha$ is the corresponding offline approximation ratio. The details are provided in Appendix D.

## 6 Experiments

In this section, we empirically evaluate our proposed algorithms on several dictionary selection problems with synthetic and real-world datasets. We use the squared $\ell^2$-utility function for all

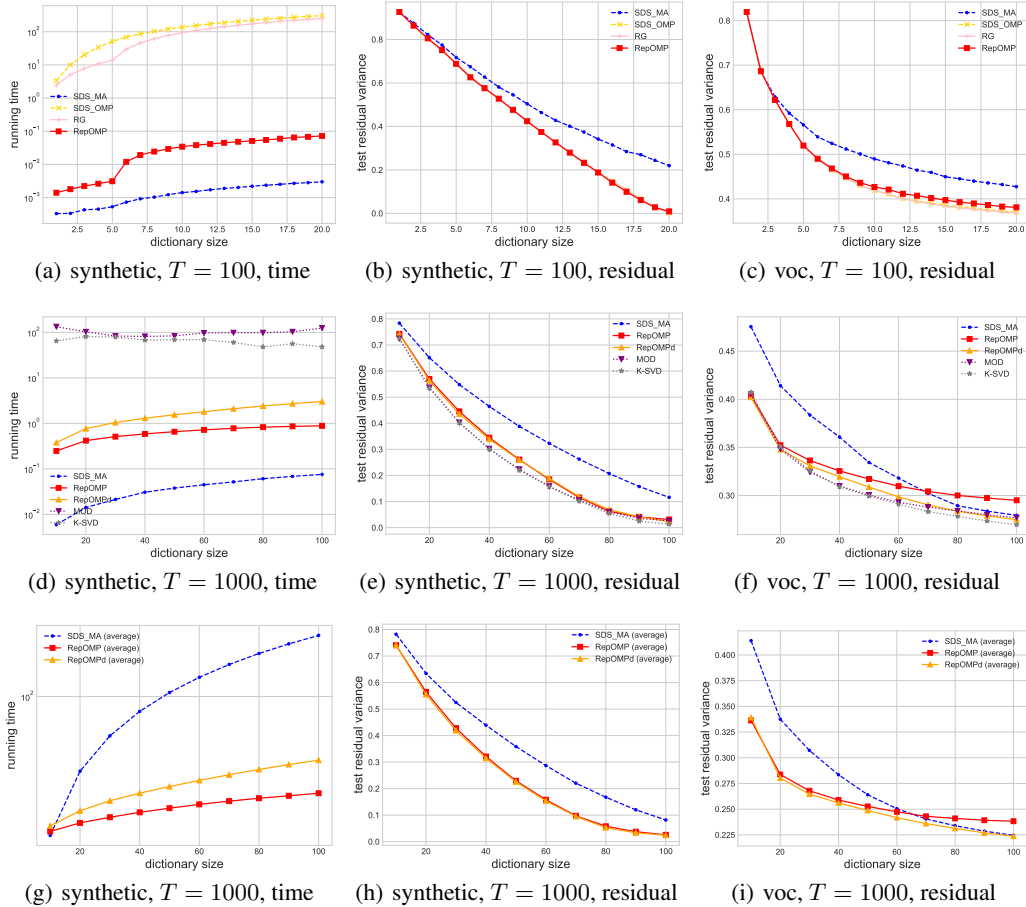

Figure 1: The experimental results for the offline setting. In all figures, the horizontal axis indicates the size of the output dictionary. (a), (b), and (c) are the results for $T = 100$. (d), (e), and (f) are the results for $T = 1000$. (g), (h), and (i) are the results for $T = 1000$ with an average sparsity constraint. For each setting, we provide the plot of the running time for the synthetic dataset, test residual variance for the synthetic dataset, and test residual variance for VOC2006 image dataset.

of the experiments. Since evaluating the value of the objective function is NP-hard, we plot the approximated residual variance obtained by orthogonal matching pursuit.

**Ground set**   We use the ground set consisting of several orthonormal bases that are standard choices in signal and image processing, such as 2D discrete cosine transform and several 2D discrete wavelet transforms (Haar, Daubechies 4, and coiflet). In all of the experiments, the dimension is set to $d = 64$, which corresponds to images of size $8 \times 8$ pixels. The size of the ground set is $n = 256$.

**Machine**   All the algorithms are implemented in Python 3.6. We conduct the experiments in a machine with Intel Xeon E3-1225 V2 (3.20 GHz and 4 cores) and 16 GB RAM.

**Datasets**   We conduct experiments on two types of datasets. The first one is a synthetic dataset. In each trial, we randomly pick a dictionary with size $k$ out of the ground set, and generate sparse linear combinations of the columns of this dictionary. The weights of the linear combinations are generated from the standard normal distribution. The second one is a dataset of real-world images extracted from PASCAL VOC2006 image datasets [15]. In each trial, we randomly select an image out of 2618 images and divide it into patches of $8 \times 8$ pixels, then select $T$ patches uniformly at random. All the patches are normalized to zero mean and unit variance. We make datasets for training and test in the same way, and use the training dataset for obtaining a dictionary and the test dataset for measuring the quality of the output dictionary.

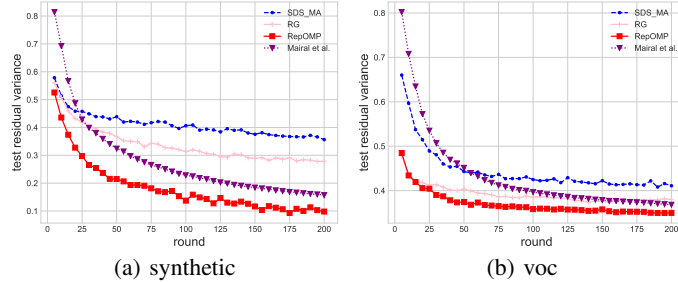

Figure 2: The experimental results for the online setting. In both figures, the horizontal axis indicates the number of rounds. (a) is the result with synthetic datasets, and (b) is the result with VOC2006 image datasets.

## 6.1 Experiments on the offline setting

We implement our proposed methods, Replacement Greedy (RG) and Replacement OMP (RepOMP), as well as the existing methods for dictionary selection, $SDS_{MA}$ and $SDS_{OMP}$. We also implement a heuristically modified version of RepOMP, which we call RepOMPd. In RepOMPd, we replace $M_{s,2}$ with some parameter that decreases as the size of the current dictionary grows, which prevents the gains of all the atoms from being zero. Here we use $M_{s,2}/\sqrt{i}$ as the decreasing parameter where $i$ is the number of iterations so far. In addition, we compare these methods with standard methods for dictionary learning, MOD [14] and KSVD [2], which is set to stop when the change of the objective value becomes no more than $10^{-6}$ or 200 iterations are finished. Orthogonal matching pursuit is used as a subroutine in both methods.

First, we compare the methods for dictionary selection with small datasets of $T = 100$. The parameter of sparsity constraints is set to $s = 5$. The results averaged over 20 trials are shown in Figure 1(a), (b), and (c). The plot of the running time for VOC2006 datasets is omitted as it is much similar to that for synthetic datasets. In terms of running time, $SDS_{MA}$ is the fastest, but the quality of the output dictionary is unsatisfactory. RepOMP is several magnitudes faster than $SDS_{OMP}$ and RG, but its quality is almost the same with $SDS_{OMP}$ and RG. In Figure 1(b), test residual variance of $SDS_{OMP}$, RG, and RepOMP are overlapped, and in Figure 1(c), test residual variance of RepOMP is slightly worse than that of $SDS_{OMP}$ and RG. From these results, we can conclude that RepOMP is by far the most practical method for dictionary selection.

Next we compare the dictionary selection methods with the dictionary learning methods with larger datasets of $T = 1000$. $SDS_{OMP}$ and RG are omitted because they are too slow to be applied to datasets of this size. The results averaged over 20 trials are shown in Figure 1(d), (e), and (f). In terms of running time, RepOMP and RepOMPd are much faster than MOD and KSVD, but their performances are competitive with MOD and KSVD.

Finally, we conduct experiments with the average sparsity constraints. We compare RepOMP and RepOMPd with Algorithm 2 in Appendix C with a variant of $SDS_{MA}$ proposed for average sparsity in Cevher and Krause [6]. The parameters of constraints are set to $s_t = 8$ for all $t \in [T]$ and $s' = 5T$. The results averaged over 20 trials are shown in Figure 1(g), (h), and (i). RepOMP and RepOMPd outperform $SDS_{MA}$ both in running time and quality of the output.

In Appendix E, We provide further experimental results. There we provide examples of image restoration, in which the average sparsity works better than the standard dictionary selection.

## 6.2 Experiments on the online setting

Here we give the experimental results on the online setting. We implement the online version of $SDS_{MA}$, RG and RepOMP, as well as an online dictionary learning algorithm proposed by Mairal et al. [24]. For all the online dictionary selection methods, the hedge algorithm is used as the subroutines. The parameters are set to $k = 20$ and $s = 5$. The results averaged over 50 trials are shown in Figure 2(a), (b). For both datasets, Online RepOMP shows a better performance than Online $SDS_{MA}$, Online RG, and the online dictionary learning algorithm.

## Acknowledgement

The authors would thank Taihei Oki and Nobutaka Shimizu for their stimulating discussions. K.F. was supported by JSPS KAKENHI Grant Number JP 18J12405. T.S. was supported by ACT-I, JST. This work was supported by JST CREST, Grant Number JPMJCR14D2, Japan.

## Footnotes

[1] A set family $\mathcal{I}$ is said to be down-closed if $X \in \mathcal{I}$ and $Y \subseteq X$ then $Y \in \mathcal{I}$.

[2] A *matroid* is a pair of a finite ground set $V$ and a non-empty down-closed family $\mathcal{I} \subseteq 2^V$ that satisfy that for all $Z, Z' \in \mathcal{I}$ with $|Z| < |Z'|$, there is an element $a \in Z' \setminus Z$ such that $Z \cup \{a\} \in \mathcal{I}$

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
