[Reviews · NeurIPS 2018]

Reviewer 1



This paper studies the dictionary selection problem. Given a large dictionary, the goal is to approximate many points using a small subset of atoms from the dictionary. The authors analyze the previous Replacement Greedy algorithm for dictionary selection objectives and then propose a variant called Replacement OMP. They show that Replacement OMP achieves comparable performance to dictionary selection/learning methods on standard datasets. They also provide extensions to an online problem and generalized structured sparsity constraints. Quality/Originality: This paper builds on existing work in an interesting and novel way. Proofs appear to be correct. Significance: This paper develops new theory to prove guarantees on dictionary selection, which improves on previous work from several years ago. Clarity: Generally a pleasure to read, but I notice that on line 50: "Replacement OMP have" Questions: How would the guarantee of Theorem 4.3 differ for Replacement Greedy? Is there any intuition for when the performance of the 2 proposed algorithms would differ? --------- EDIT: After reading the author response, I feel that they have addressed my minor question about Theorem 4.3. They also addressed the concerns of other reviewers. Therefore I am keeping my review as clear accept.

Reviewer 2



This paper studies the problem of dictionary selection, where the goal is to pick k vectors among a collection of n d-dimensional vectors such that these vectors can approximate T data points in a sparse representation. This problem is well-studied and the authors propose a new algorithm with theoretical guarantees which is faster than previous algorithms and which can handle more general constraints. This algorithm is based on a previous algorithm for the related problem of two stage submodular maximization called replacement greedy. It is first shown that replacement greedy also enjoys approximation guarantees for dictionary selection. Then, the authors further improve this algorithm to obtain replacement OMP, which is faster. This algorithm is also extended to the online setting. The authors evaluate their algorithm empirically and compare it to previous algorithms for both dictionary selection and dictionary learning (the continuous version of the problem). It is shown that replacement OMP is either significantly faster with similar performance or has better performance than any benchmark. Overall, this paper makes a solid contribution to the well-motivated and well-studied problem of dictionary selection. Previous algorithms had very slow running time, so the significant improvement in running time is important. The techniques are nice and port recent ideas from the related problem of two stage submodular maximization (it is not straightforward that these techniques can be used for this problem with approximation guarantees), and the authors also give non-trivial further improvements to the algorithm by using a nice idea of orthogonal matching pursuit. Another important contribution is that this algorithm can be used under generalized sparsity constraints, which are well-motivated and relevant to the problem studied. The approximation guarantees obtained in this paper are incomparable to those from previous papers, but the experiments clearly demonstrate that the proposed algorithm either performs comparably to previous algorithms which are much slower or outperforms previous algorithms which are faster. I thought it was nice that the authors gave a comparison of discrete and continuous methods, showing a significant improvement in the running time compared to the algorithms from the continuous setting for dictionary learning. Minor cons: - the empirical performance of SDSma does not seem as “poor” as the paper says. - it would be nice if the authors discussed the approximation guarantees obtained in the different experimental settings, since these approximations are incomparable to those of other algorithms in general. In other words, what are the constants ms, Ms, Ms2,… for the datasets used for the experiments?

Reviewer 3



This paper proposes an algorithm for the dictionary selection problem. In this problem the goal is to find a small subset of "atoms" so it possible to approximate the whole data using the selected subset in sparse representation. The proposed algorithm is an extension to "Replacement Greedy", an algorithm for two-stage submodular maximization. This is a similar problem to the paper's problem, however the underline function in this paper is not submodular. The algorithm uses techniques in Orthogonal matching pursuit (OMP). This technique allows them to provably improve the running time of the algorithm and be able to pose global constraints. Experimental results show improvement in runtime and tests residual variance in comparison with other existing algorithms. The technical parts appear to be correct. -In the experiment section, the author used test residual variance to compare between the algorithms. I don’t understand why? Isn't it better comparing the value of the computed subset? After all the goal is to maximize this function. In my opinion, I think the authors should at least explain their choice. -The algorithm requires the smooth parameter M_{s,2}, this parameter can be bounded. According to the author, this value can be computed in O(n^2 * d) time. This sounds much higher than the run time of this algorithm and other existing algorithms. What am I missing? -The paper is generally well written and constructed. There was one notation that I think could at least be explained better- equation (5) the \_Z_t'\Z_t. It took me a while to understand it. -The p-replacement definition was a little bit confusing. At first I thought it is just a chain of replacement, that in each step approaches (Z_1*...Z_t*). It is possible to write a small explanation before the definition. -An intuition for line 179-180 is needed. "we propose a proxy of this quantity by borrowing an idea from orthogonal matching pursuit". -The main contribution of this paper, in my opinion, is the extension of OMP, which gives a better runtime and the possibility to pose global constraints. -The paper also proposes an online variant of their algorithm. This variant is the end of the supplementary, which I didn’t fully verify. -It seems the paper has shown better results on the dictionary selection problem both in theory and application. Other comments and typos: -Table 1: The approximation ratio of the replacement greedy is from your paper. I think it should be stated somewhere in the caption. -Typo Eq(3)- You should replace the big T with small t: Z_t' \subset Z_t+a. -Line 370 in the supplementary: "there exists a bijection"- what is the properties of this bijection? Maybe it is clear from the context but I still think you should add it. ---Edit---- I have read the author response. The clarification about the upper bounds of M_{s,2} satisfied me (the second comment). Therefore, I am updating my rating of this paper from 6 to 7.